# Changes in Meat of Hu Sheep during Postmortem Aging Based on ACQUITY UPLC I-Class Plus/VION IMS QTof

**DOI:** 10.3390/foods13010174

**Published:** 2024-01-04

**Authors:** Jie Xu, Qiang Wang, Yimeng Wang, Menghuan Bao, Xiaomei Sun, Yongjun Li

**Affiliations:** College of Animal Science and Technology, Yangzhou University, Yangzhou 225009, China; 15150815661@163.com (J.X.); junyouwang26@gmail.com (Q.W.); mx120220861@stu.yzu.edu.cn (Y.W.); 13357160331@163.com (M.B.); yzliyongjun@163.com (Y.L.)

**Keywords:** *longissimus dorsi* (LD), postmortem aging, muscle fibers, ACQUITY UPLC I-Class Plus/VION IMS QTof

## Abstract

Meat and meat products have a critical role in the human diet as important high-nutrient foods that are widely consumed worldwide. This study evaluated the effects of postmortem aging on Hu sheep’s meat quality in the *longissimus dorsi* (LD) muscle during postmortem aging. The samples were stored at 4 ± 1 °C; the meat quality was measured at 6 h, 12 h, 24 h, 36 h, 48 h, 72 h, 96 h, 120 h, 144 h, and 168 h of postmortem aging. The results showed that, during the postmortem aging process, the pH of the muscles first decreased and then increased, and the shear force first increased and then decreased. The muscle fiber skeleton began to degrade, and the overall meat quality was improved to some extent. In addition, through ACQUITY UPLC I-Class Plus IMS Qtof identification of the muscle samples at different time points during the postmortem maturation process of the meat of Hu sheep, a total of 2168 metabolites were identified, and 470 metabolites were screened based on the VIP, P, and FC values, of which 79 were involved in KEGG pathways. In addition, pathways such as sphingolipid metabolism, glycerophospholipid metabolism, phenylalanine metabolism, and fatty acid elongation and degradation play an important role in the metabolic product changes in the meat of Hu sheep throughout the entire maturation process. These findings provide some insights into the changes in meat quality during the post-slaughter maturation process of lake lamb.

## 1. Introduction

Meat and meat products have a critical role in the human diet as important high-nutrient foods that are widely consumed worldwide [1]. Due to the current increasing demand for meat, China’s meat market may become the world’s largest meat market in the future [2]. At the same time, lamb is becoming increasingly popular in China and has become an important component of residents’ diets [3]. Consumers believe that tenderness and flavor can affect the edible quality of lamb. In addition, consumer demand for food is shifting towards high quality. At present, the meat industry is committed to quantifying high-quality meat products to meet the needs of consumers [4].

The quality of meat is influenced by many factors. Previous studies have shown that meat quality may be affected by age [5], muscle fibers [6], and postmortem aging [7]. Among these, postmortem aging ultimately has the greatest impact on meat quality [7]. Meat aging refers to the storage of meat at a low temperature (4 ± 1 °C) to improve the quality of various meat traits [8]. Many complex processes occur postmortem in all carcass muscles, resulting in increased meat tenderness and a desirable flavor and aroma profile. Ijaz et al. [9] found that as the storage time increases, the shear strength of beef decreases, and the meat color is enhanced. Orkusz et al. [10] identified the lipid oxidation and color changes in goose meat stored under aging. Hwang and Hong [11] found that wet aging is beneficial for producing tender and juicy pork, and moderate pressure (150 Mpa) can improve the quality of wet-aged pork.

To our knowledge, the effects of postmortem aging on meat quality in Hu sheep have not been studied. However, the Hu sheep is a special meat breed in Jiaxing or the Taihu Lake area of Zhejiang Province [12]. Different from normal meat-producing goats, the Hu sheep, as a new type of meat sheep, is very popular in the consumer market of China. Compared to other sheep breeds, Hu sheep mature earlier, produce more lambs per litter, and are in heat throughout the year, making them an advantageous breed for overall lamb production. In addition, the intramuscular fat and fatty acid contents of lake lamb are relatively high [13], which may lead to their involvement in enhanced lipid metabolism during the post-slaughter maturation process, thereby affecting the flavor of the meat. Therefore, it is necessary to measure the changes in basic meat quality and metabolites during the meat’s maturation.

ACQUITY UPLC I-Class Plus/VION IMS QTOF consists of a mass spectrometry analysis system called the ACQUITY UPLC I-Class Plus, which is used for difficult research, and a VION IMS QTOF that can perform high-resolution mass spectrometry analysis. The resolution of this instrument is greater than 50,000 FWHM, with a mass accuracy of less than 1 ppm, and it has significant advantages in metabolite identification [14]. Here, we studied the longest dorsal muscle of Hu sheep using this instrument. The purpose of this study is to investigate the changes in the basal meat quality and metabolites of the *longissimus dorsi* muscles of Hu sheep during postmortem maturation, and to identify the appropriate maturation time and unique metabolites and pathways of Hu sheep muscles.

## 2. Materials and Methods

### 2.1. Animals

The experimental sheep were selected from Qianbao Animal Husbandry in Jiangsu Province. Eight 8-month-old male Hu sheep weighing around 40 kg were selected, with similar feeding and management modes and good health conditions. All animal procedures were conducted in accordance with the regulations of the Experimental Animal Ethics Committee of Yangzhou University (No. 202206132). The selected sheep were fasted for 24 h before slaughter and then transported to a slaughterhouse and exsanguinated after being stunned. The carcasses were placed in a scald tank and dehaired at 65 °C; then, a knife was used to remove the remaining hair. After slaughter, the longest dorsal muscle was taken from the posterior edge of the last rib at a distance of 3 cm from the midline of the back, vacuum-packaged, and refrigerated at 0–4 °C.

### 2.2. The Quality Characteristics

#### 2.2.1. pH

The pH value was determined according to the method of Huang et al. [15] with slight modifications. Meat samples were collected at 11 time points, including 0 h, 6 h, 12 h, 24 h, 36 h, 48 h, 72 h, 96 h, 120 h, 144 h, and 168 h. A portable pH meter (testo205; Guangzhou, China) was used for the pH measurement at each time point. The probe was cleaned with ultrapure water, and the pH meter was calibrated before the measurement. When measuring, the probe was completely in contact with the meat sample, avoiding the fat and tendons. The experiment was repeated three times, and the average value was taken.

#### 2.2.2. Shear Forces

The shear forces of the samples were measured using the method established by Hou et al. [16]. Meat samples were collected at 10 time points, including 6 h, 12 h, 24 h, 36 h, 48 h, 72 h, 96 h, 120 h, 144 h, and 168 h. The surface fat, tendons, and membranes were removed. The meat samples were placed in a polyethylene bag and heated in a 75 °C water bath until their center temperature was 70 °C; then, the samples were taken out and left to cool to room temperature. A meat tenderness tester (RH-N50; Guangzhou, China) was used to measure the shear force values of the samples. The number of samples was no less than 6, and the average value was taken after removing abnormal values.

#### 2.2.3. Loss Rate

A meat strain hydraulic tester was used to measure the water loss rate of the meat samples (RH-1000; Guangzhou, China). The mass of the muscle before compression was recorded. The meat sample was placed flat in the middle of a piece of filter paper, and the filter paper containing the meat sample was moved to the table of the testing base to start the test. After compression, the sample was taken out, the mass after compression was immediately weighed, and the water loss rate was calculated. The weight of the meat before compression is denoted as A, and after compression, it is denoted as B. The formula for the loss rate is 100% × [(A − B)/A].

#### 2.2.4. Color Measurements

Meat samples were collected at 9 time points, including 0 h, 6 h, 12 h, 24 h, 36 h, 48 h, 72 h, 120 h and 168 h. A portable color difference instrument (CR-400, Konika Minolta, Tokyo, Japan) was used to measure the color of the meat samples. The measurement aperture of the CR400 is 8 mm, the light source is C, and the color difference reference color is 100. Cross-sections of the meat samples were cut and flattened so that they were parallel to the tabletop. The lens of the color difference meter was placed perpendicular to the surface of the meat sample, with the lens hole positioned close to it. The values of L*, a*, and b* were recorded at 3–4 sites. Abnormal values were removed, and the average value was taken. The parameters for determining color were L*, a*, and b*.

#### 2.2.5. TVB-N

TVB-N was measured using the automatic Kjeldahl nitrogen analyzer (FOSS8400, Foss Company, Beijing, China) method. The experimental method was based on the national food safety standard for the determination of volatile basic nitrogen in food (GB 5009.228-2016 [17]). After homogenizing the sample, 10 g was weighed, accurate to 0.001 g, and placed in a distillation tube. Then, 75 mL of water was added, and the tube was shaken so that the sample was evenly dispersed in the solution. The sample was immersed for 30 min. Afterwards, 1 g of magnesium oxide was added, and the sample was immediately connected to a still. The measurement was started according to the conditions set for the instrument and the requirements of the instrument operating manual. The experiment was repeated three times, and the average value was taken.

The content of volatile base nitrogen in the sample was calculated according to the formula X = (V1 − V2) × C × 14/m × 100.

V1 is the volume of hydrochloric acid or sulfuric acid standard titration solution consumed by the test solution, mL; V2 is the volume of hydrochloric acid or sulfuric acid standard titration solution consumed by the reagent blank, mL; C is the concentration of hydrochloric acid or sulfuric acid standard titration solution, mol/L; 14 indicates that the mass of nitrogen equivalent is titrated to 1.0 mL HCl [c(HCl) = 1.00 mol/L], g/mol; M is the sample mass, g; and 100 indicates that the calculation result is converted to mg/100 g.

### 2.3. Observation of Muscle Fiber Morphology

#### 2.3.1. HE Staining and Observation

Using the hematoxylin and eosin staining method [18], the sections for staining were selected, and then dewaxing and alcoholization, staining, dehydration, transparency, and sealing were performed in sequence. The prepared glass slides were observed under an upright microscope, and the morphology of the muscle fibers was photographed under a 200× oil microscope.

#### 2.3.2. Staining and Observation of Skeleton Proteins

Embedded tissue was collected and prepared into frozen sections [19], which were stored at −20 °C. Fluorescence staining was performed using a ghost pen cyclic peptide [20], and during subsequent storage, the sections were kept away from light at 4 °C.

The prepared glass slides were observed under an ultrahigh-resolution laser confocal microscope, and the shape of the muscle fiber skeleton proteins was photographed under a 200× oil microscope.

### 2.4. Metabolite Extraction from Hu Sheep Meats

The extraction of metabolites was carried out using the method described by Jia et al. [21]. After homogenizing the meat sample, it was subjected to vortex and ice bath ultrasonic treatment, centrifuged at 10,000× *g* at 4 °C for 20 min, and then transferred the supernatant for a nitrogen-purging instrument. Before liquid chromatography separation, reconstitution and filtration with 0.22 μm membranes was performed.

### 2.5. UPLC I-Class Plus Method

Untargeted metabolomics profiling was conducted using an ACQUITY UPLC I-Class Plus (PDA eλ Detector)-VION IMS Qtof equipped with an electric spray source (ESI) and atmospheric pressure chemical source (APCI) composite ion source, which operated in positive ion mode. The instrument parameters for this method followed those established by [21].

### 2.6. Data Processing and Statistical Analysis

QI 2.4 metabolite analysis software was used for peak comparison, retention time correction, and peak area extraction of the raw data. The data extracted using QI 2.4 software were first subjected to metabolic product structure identification, data preprocessing, experimental data quality evaluation, and, finally, data analysis. After preprocessing the dataset, data such as the relative peak area, retention time, ion mode, and mass charge ratio of the identified metabolites were obtained. They were imported into SIMCA 14.0 software for partial least squares discriminant analysis for pattern recognition analysis to screen differential metabolites. MetaboAnalyst 5.0 was used to perform differential metabolite hierarchical clustering analysis, differential metabolite correlation analysis, and KEGG pathway analysis on the relative peak area of the differential metabolites.

## 3. Results and Discussion

### 3.1. pH and Share Forces

As the maturation time after death was prolonged, the pH value of the longest dorsal muscle of the Hu sheep first decreased and then increased (Figure 1). At 0–6 h, the pH significantly decreased. At 6–12 h, the pH significantly decreased. At 12–168 h, the pH slowly increased (Figure 1). The reason for this phenomenon is that after slaughter, the oxygen in the body of Hu sheep is interrupted, anaerobic respiration occurs in the body, glycogen is decomposed into lactic acid, and adenosine triphosphate is decomposed into phosphate, resulting in a significant decrease in the pH value of Hu sheep (*p* < 0.05). Afterwards, with the extension of the oxidation time, lactic acid gradually degrades, the protein in the meat is decomposed into alkaline substances such as ammonia and amino acids, meaning that the meat tends to be alkaline, and the pH value significantly increases [22].

The tenderness, also known as softness, of meat is one of the important elements that make up meat quality, and it is mainly reflected in its taste. The morphology of connective tissue in muscles, the degree of protein hydrolysis in muscles, and the structure of muscle fibers affect the tenderness of meat. The shear force value directly reflects the tenderness of meat, and the smaller the index within a certain range, the more tender the meat. From Figure 2, it can be seen that during the postmortem maturation process, the overall shear force value of the meat showed a trend of first increasing and then decreasing. At 6–12 h, the shear force of the meat significantly increased, reaching a maximum value of 69.63 N at the 12th hour (Figure 2). This may be due to the shortening of muscle fibers in the early stages of maturity, leading to hardening of the meat, which enters a stiff state. During the 12–24 h period, although the shear force decreased slightly, it was still higher than that at 6 h. Based on previous studies [23] and the continuous decrease in the pH during this period, it is speculated that there is a completely rigid time point between 12 and 24 h in this study. As the postmortem ripening time is prolonged, the meat begins to enter the thawing and ripening period. From Figure 2, it can be seen that, compared with the shear force at 12 h, the shear force at 36 h was significantly reduced. Compared with the shear force at 36 h, the shear force at 72 h was significantly reduced. Compared with the shear force at 72 h, the shear force at 144 h was significantly reduced. This is because during the maturation process, the contraction state of sarcomeres changes, and the shear force gradually decreases.

From Figure 1 and Figure 2, it can be seen that while the pH significantly decreased from 0 to 12 h, the shear force significantly increased, and at the same time, as the pH slowly increased from 12 to 168 h, the shear force decreased. This is because after slaughter, aerobic respiration is converted into anaerobic respiration, glycogen is decomposed into lactic acid, and the pH value of Hu sheep significantly decreases (*p* < 0.05). At the same time, the ATP content sharply decreases, causing the sarcoplasmic reticulum to self-disintegrate, and a large amount of calcium ions rush into the myofibrils, forming a large amount of the irreversible AM complex, leading to rapid muscle contraction and a decrease of approximately 60% in muscle elongation, resulting in the disappearance of muscle softness. The carcass appears stiff and straight. Afterwards, as the oxidation time is prolonged, the protein in the meat decomposes, and the pH increases. At the same time, the state of the actin complex is released as the protein decomposes, resulting in a decrease in shear force.

### 3.2. Color, Loss Rate, and TVB-N

The change in flesh color is mainly determined by the binding of myoglobin with oxygen [10], which is divided into oxygenated myoglobin (MbO_2_), deoxygenated myoglobin (Mb), and metmyoglobin (MMb). The three transform each other, resulting in muscles presenting different colors, such as dark purple, bright red, and brown. The higher the L* value, the brighter and whiter the color. The a* value represents a positive timing, indicating a slight redness, and the b* value represents a positive timing, indicating a yellowish color. At 12–72 h after slaughter, there was a significant increase in the meat’s a* (Figure 3) and b* (Figure 4) values, which may have been due to the oxidation of Mb which produced MbO_2_, resulting in an increase in the relative content of MbO_2_ and a bright red color of the meat. From Figure 5, it can be seen that the L* value of the meat showed a trend of first increasing and then decreasing. After slaughter, the L* value significantly increased from 0 to 12 h, which may have been due to surface water accumulation during the postmortem ripening process of the meat, enhancing its ability to reflect light.

The loss rate is one of the important indicators for measuring the quality of meat, reflecting the water retention of meat to a certain extent. The water retention of meat directly affects the color, tenderness, and nutritional composition of meat products [21]. The loss rate of meat increased significantly (Figure 6) (*p* < 0.05). As the aging time is prolonged, the water loss rate of meat continually increases. This is because during the aging process, the myofibrillar protein structure of meat is damaged due to protein degradation, resulting in a decrease in its water retention ability and an increase in its water loss rate. The protein in the meat denatures and contracts, causing the structure of myofibrillar proteins to be disrupted, resulting in a decrease in the water retention capacity of the meat and an increase in cooking losses.

The TVB-N value is an important index for evaluating meat freshness [21]. With the increase in the oxidation time, the content of TVB-N in the meat increased significantly (Figure 7). Prior to 96 h, the TVB-N content of the meat samples at all time points was less than 15 mg/100 g, meaning that the samples could be classified as first-class fresh meat. Starting from 96 h, the TVB-N content of some meat samples exceeded 15 mg/100 g, and after 168 h, this value was exceeded in all of the meat samples, belonging to the category of second-class fresh meat. From 6 to 72 h, due to protein hydrolysis, a certain amount of basic nitrogen-containing substances were dissociated, resulting in a gradual increase in TVB-N. From 96 to 168 h, as the postmortem maturation time was prolonged, the microorganisms on the meat surface began to proliferate in large quantities, resulting in the production of a large amount of basic nitrogen compounds, leading to a gradual increase in the TVB-N content.

### 3.3. Muscle Fibers

From the cross-sectional analysis of the muscle fibers shown in Figure 8, it can be concluded that at 6 h after slaughter, the muscle tissue was very tight, and the cell gap was very small. At 12 h after slaughter, the area of muscle fibers began to decrease, and the intercellular space began to increase. Meanwhile, 24 h after slaughter, the muscle fibers began to shrink, the diameter of the muscle fibers gradually decreased, and the intercellular space significantly increased, but some muscle fibers began to swell and become larger. From the 48th hour after slaughter, we can see that some muscle cell centers became hollow, and the muscle cells were significantly damaged. At the same time, some muscle fibers disappeared, and the original areas turned white.

From Figure 9, it can be seen that the skeletal protein of muscle cells was intact at 9 h after slaughter, showing a clear contrast between the light and dark bands and the linear shape of the muscle fibers. As the postmortem maturation time was prolonged, the muscle fibers began to distort 12 h after slaughter. At 24 h after slaughter, the muscle fibers started to become swollen and enlarged, which is consistent with the results of the cross-sectional view of the muscle fibers. Swelling and hypertrophy of muscle fibers are likely manifestations of an inflammatory response. At 48 h after slaughter, the muscle fibers were severely distorted, and some began to break. At 72 h after slaughter, almost all of the muscle fibers were broken, indicating that the tenderness of the meat was better at this time, which is consistent with the significant decrease in the shear strength of the meat between 24 and 72 h after slaughter.

### 3.4. Untargeted Metabolic Profiling of Hu Sheep

To comprehensively identify the metabolites of Hu sheep meat, untargeted metabolic profiling was performed using an ACQUITY UPLC I-Class Plus IMS QTof on six sample groups, namely 6 h, 12 h, 24 h, 48 h, 72 h, and 96 h, which generated 3196 ion features in the ESI and APCI composite ion source (+). Finally, a total of 2168 metabolites were commonly characterized in the meat, and after removing duplicates, a total of 470 differential metabolites were screened (Appendix A), of which 79 were involved in KEGG pathways.

### 3.5. Comparison of Metabolites of Hu Sheep Meat

In order to understand the changes in different metabolic products during the postmortem aging process of Hu sheep meat, we used PLS-DA for analysis using SIMCA14.0. PLS-DA is a common pattern recognition method used to analyze the overall distribution of meat metabolites at different aging times. As shown in the PLS-DA scoring chart (Figure 10), six meat samples were distinguished based on the first two main components (t1 and t2). According to PLS-DA, the six groups of meat samples could be divided into two groups: one lasting from 6 h to 24 h, and the other lasting from 48 h to 96 h. This indicates that during the aging process of meat, significant changes in metabolites occur from 48 h onwards. In addition, QC was clustered in various groups, demonstrating the high repeatability and stability of the analytical instruments.

A permutation test was conducted on the above PLS-DA results. As displayed in Figure 11, the values of the classification parameters R2Y and Q2 were 0.8536 and 0.7637, respectively, indicating that there was a good fitting and predictive power. A permutation test with 200 iterations showed that the intercept values of R2 and Q2 were (0, 0.173) and (0, −0.31) in positive mode (Figure 11). All blue Q2 values to the left are lower than the original points to the right, and the blue regression line of the Q2 points intersects the vertical axis (on the left) at or below zero. All green R2 values to the left are lower than the original points to the right, thus indicating that the PLS-DA mode is reliable. Afterwards, the variable importance in the VIP of the PLS-DA results was used to determine the differential metabolites among the meat samples with different postmortem maturation times.

In this study, metabolites with VIP > 1, *p* < 0.05, and FC > 2 or FC < 0.5 were regarded as differential metabolites. In the above statistical analysis, 470 critical metabolites determined in positive mode were responsible for the metabolic changes during the postmortem ripening of the meat. Out of the 470 metabolites, 79 could be matched with relevant KEGG IDs (Table 1). The heat maps of the meat samples and 79 metabolites are shown in Figure 12. Through observation, it can be seen that the samples can be separated into three clusters. Cluster I includes 6 and 12 h, Cluster II includes 24 and 48 h, and Cluster III includes 72 and 96 h, which is highly consistent with the results of the PLS-DA model.

### 3.6. Metabolic Pathway Analysis

To explore the most important pathways related to the metabolic responses of meat samples with different aging times, the differential metabolites among the six samples were subjected to pathway analysis using MetaboAnalyst 5.0. According to Figure 13 and Table 2, 20 pathways were predicted, where 2 pathways had impact values higher than 0.1 and 3 pathways had *p* values less than 0.05. These pathways included sphingolipid metabolism, glycerophospholipid metabolism, phenylalanine metabolism, and fatty acid elongation and degradation, providing some insights into the metabolic changes during postmortem aging.

### 3.7. Differential Metabolites Related to the Flesh Quality of Hu Sheep Meat

Generally speaking, the quality of meat mainly depends on the related changes in proteins [24] and lipids [25]. Lipid oxidation may affect meat quality by affecting the formation of flavor compounds in meat [26], while proteins and amino acids may play important roles as flavor precursors [27] and may also affect the taste, tenderness, and water retention of meat [28]. In addition, the degradation of proteins and fats produces alkaline nitrogen-containing substances, leading to an increase in the TVB-N and pH [21]. Therefore, this study investigated the changes in and pathways of related metabolites during the maturation process of the longest dorsal muscle in Hu sheep.

Amino acids are important substances for maintaining metabolism and protein synthesis. According to Table 2, the main metabolic pathways related to amino acids are phenylalanine metabolism, arginine and proline metabolism, and aminoacyl tRNA biosynthesis. According to Table 1, the related metabolites are phenylacetaldehyde and L-proline. According to Figure 12, phenylacetaldehyde was discovered in the late stage of maturation, appearing during the maturation time of 72–96 h. Phenylacetaldehyde is a fatty aldehyde with a fragrance similar to hyacinth. When diluted, it has a sweet, fruity aroma and is insoluble in water. According to reports, phenylacetaldehyde appears in roasted chicken and not in steamed chicken [29], indicating that phenylacetaldehyde has unique characteristics and optimizes the flavor of meat to a certain extent. According to Figure 12, L-proline appeared during the maturation time of 48–96 h. L-proline is one of the important amino acids for synthesizing human proteins, mainly used as a nutritional supplement and flavor agent. It can undergo amino acid carbonyl reactions with sugars and generate special aroma substances. According to reports, L-proline can be found in donkey [30] and goat [31] meat, giving the meat a unique sweetness. In the purine metabolism pathway, significant changes were detected in three metabolites, namely adenosine monophosphate, hypoxantine, and inosine. Adenosine monophosphate appeared at a maturation time of 6 h and disappeared from 12 h onwards. Moreover, hypoxantine and inosine appeared from 48 h onwards. This maybe because adenosine monophosphate can be decomposed into adenosine, adenine, or IMP, leading to an increase in inosine and hypoxantine. According to reports, AMP (adenosine monophosphate) may have a certain improvement effect on the tenderness of meat, leading to more myofibrillar fragmentation and structural changes in myofibrillar proteins [32].

In meat-related research, lipid metabolism is an important link. In sphingolipid metabolism, sphinganine increased from 48 h, and phytosphingosine increased from 72 h. Sphinganine is a very hydrophobic molecule, which may lead to an increase in the water loss rate at 48 h. The emergence of phytosphingosine can improve the water retention ability to a certain extent. In addition, phytosphingosine can induce cell apoptosis [33], which may further lead to the degradation of myofibrillar cytoskeleton proteins and a decrease in shear force. In glycerophoric metabolism, four types of phenology were present, with PE (14:0/24:0) present in the early stages of maturation. PE (14:1 (9Z)/22:0), PE (15:0/22:0), and PE (15:0/22:1 (13Z)) appeared from 48 h onwards. According to reports, the formation of phosphatidylethanolamine is closely related to aldehydes [34]. In addition, starting from 48 h, an increase in five types of lysophospholipids and one type of lysophosphatidic acid was detected. According to reports, lysophospholipids significantly increase during the heating process of chicken [35]. Sodium palmate was found at 48 h, which also greatly improved the quality of meat.

## 4. Conclusions

The ACQUITY UPLC I-Class Plus IMS QTof is an instrument that can effectively identify metabolites. It is an effective method for identifying changes in metabolic products during the postmortem maturation process of lake lamb meat, and it can screen out a large number of differential metabolites. In this study, sphingolipid metabolism, glycerophospholipid metabolism, phenylalanine metabolism, and fatty acid elongation and degradation were observed to be significant metabolic pathways. Phenylacetaldehyde and L-proline enhance the flavor of meat. AMP, sphinganine, and phytosphingosine may affect the water retention and shear force values during muscle maturation. During the maturation process of meat, these metabolites may play a crucial role. This study elucidates the changes in basic meat quality and muscle fiber morphology and structure, as well as the relative changes in metabolomics and metabolic pathways during the postmortem maturation process of Hu sheep meat. Overall, these findings provide a good reference for the study of the quality of meat in Hu sheep.

## Figures and Tables

**Figure 1 foods-13-00174-f001:**
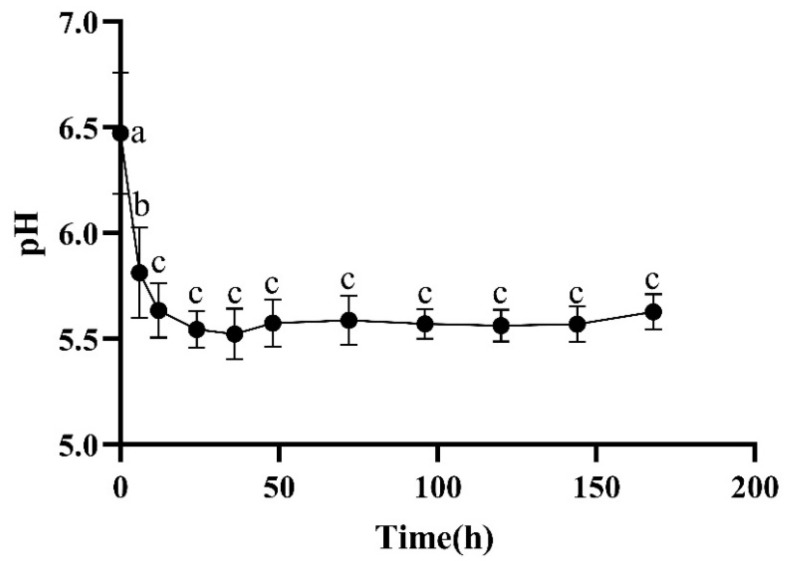
Changes in pH during postmortem maturation. Means with different letters (a–c) are significantly different (*p* < 0.05).

**Figure 2 foods-13-00174-f002:**
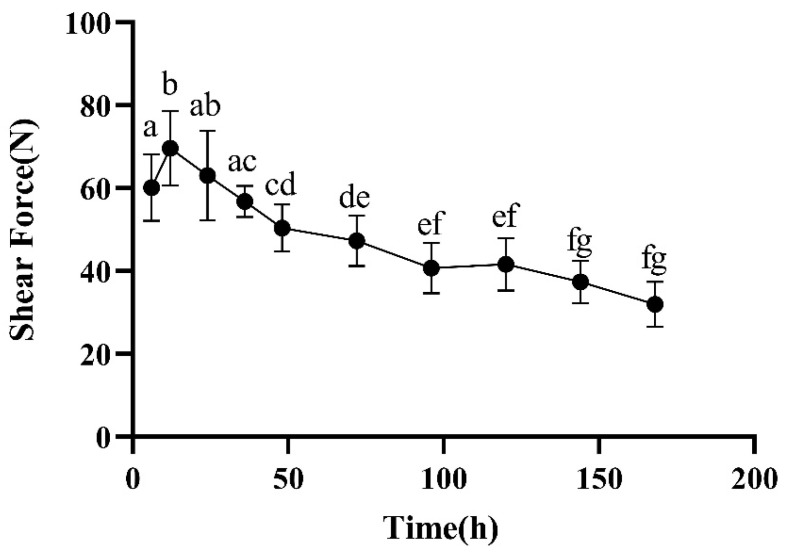
Changes in shear force during postmortem maturation. The higher the shear force, the lower the tenderness of the meat. Means with different letters (a–g) are significantly different (*p* < 0.05).

**Figure 3 foods-13-00174-f003:**
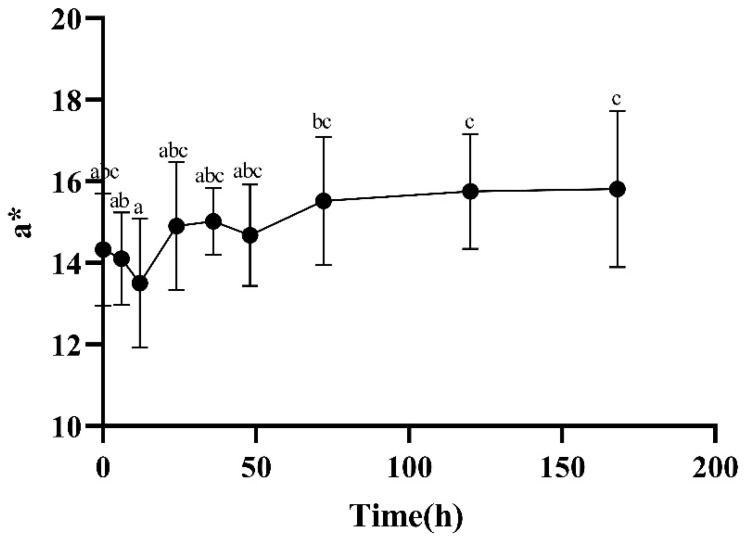
Changes in a* during postmortem maturation. Means with different letters (a–c) are significantly different (*p* < 0.05).

**Figure 4 foods-13-00174-f004:**
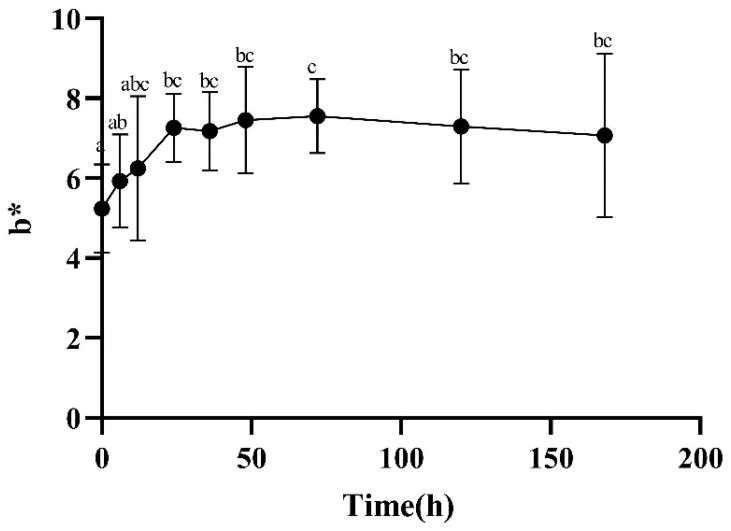
Changes in b* during postmortem maturation. Means with different letters (a–c) are significantly different (*p* < 0.05).

**Figure 5 foods-13-00174-f005:**
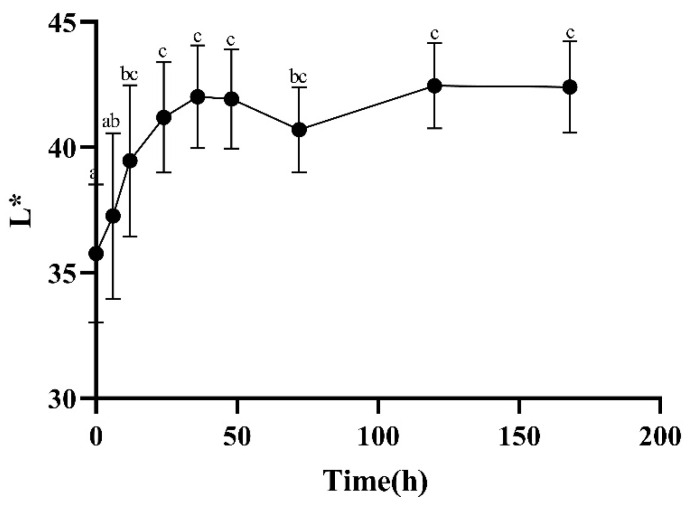
Changes in L* during postmortem maturation. Means with different letters (a–c) are significantly different (*p* < 0.05).

**Figure 6 foods-13-00174-f006:**
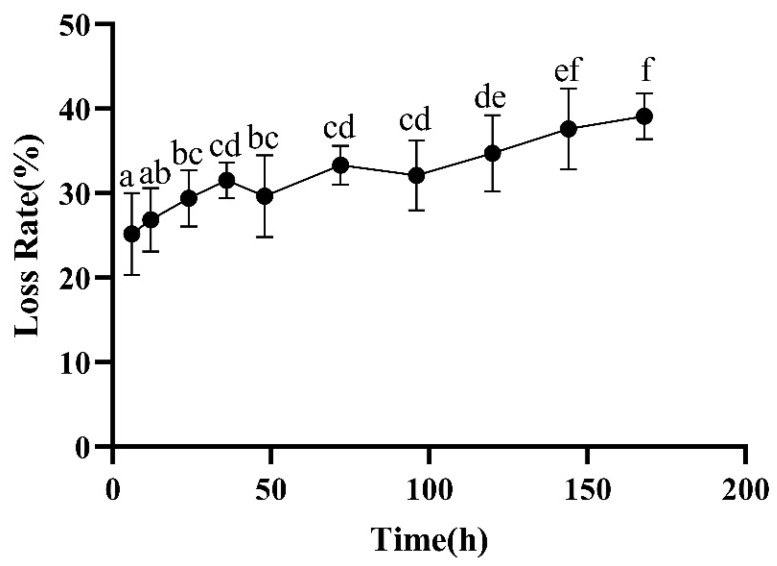
Changes in loss rate during postmortem maturation. Means with different letters (a–f) are significantly different (*p* < 0.05).

**Figure 7 foods-13-00174-f007:**
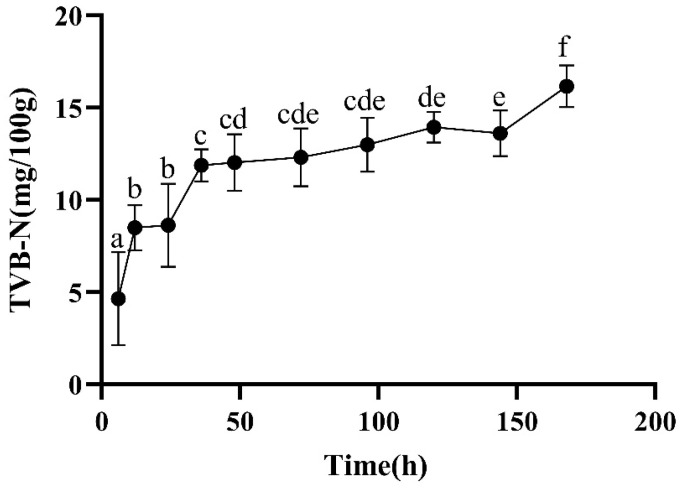
Changes in TVB-N during postmortem maturation. The higher the TVB-N content, the lower the freshness of the meat. Means with different letters (a–f) are significantly different (*p* < 0.05).

**Figure 8 foods-13-00174-f008:**
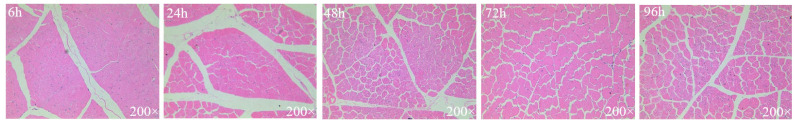
Changes in muscle fiber morphology during postmortem maturation.

**Figure 9 foods-13-00174-f009:**
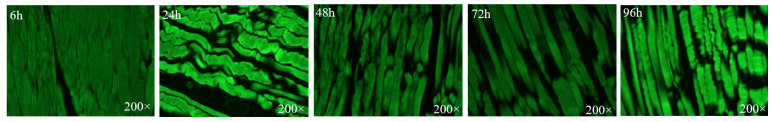
Immunofluorescence staining of muscle fiber skeleton protein.

**Figure 10 foods-13-00174-f010:**
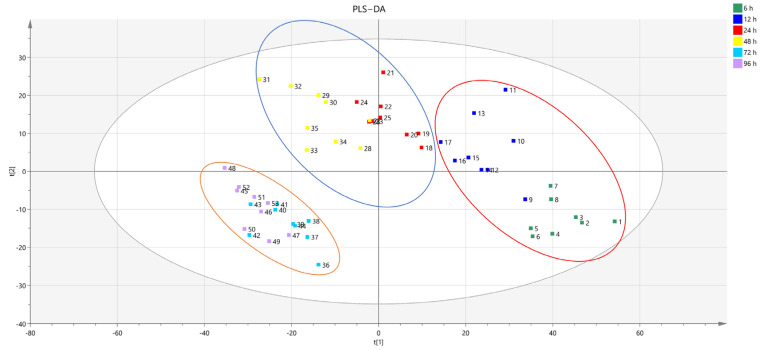
PLS-DA score plots of samples acquired in positive mode.

**Figure 11 foods-13-00174-f011:**
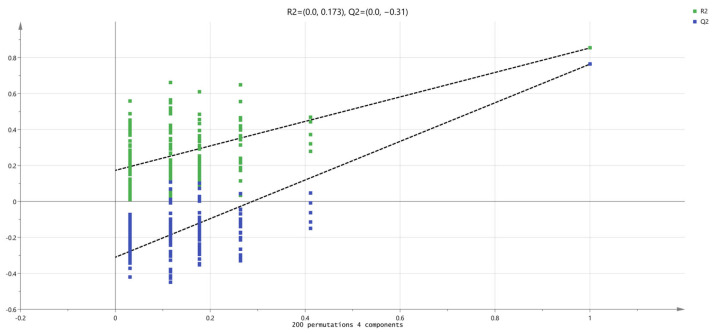
The validation of the PLS-DA model using permutation testing (200 iterations) in positive mode.

**Figure 12 foods-13-00174-f012:**
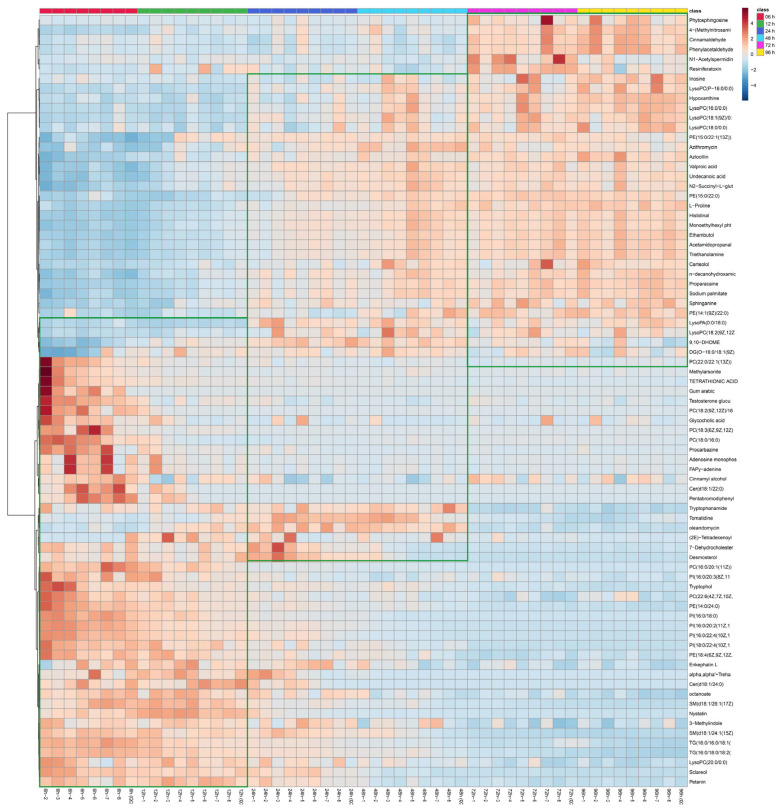
Heat map visualization of differential metabolites.

**Figure 13 foods-13-00174-f013:**
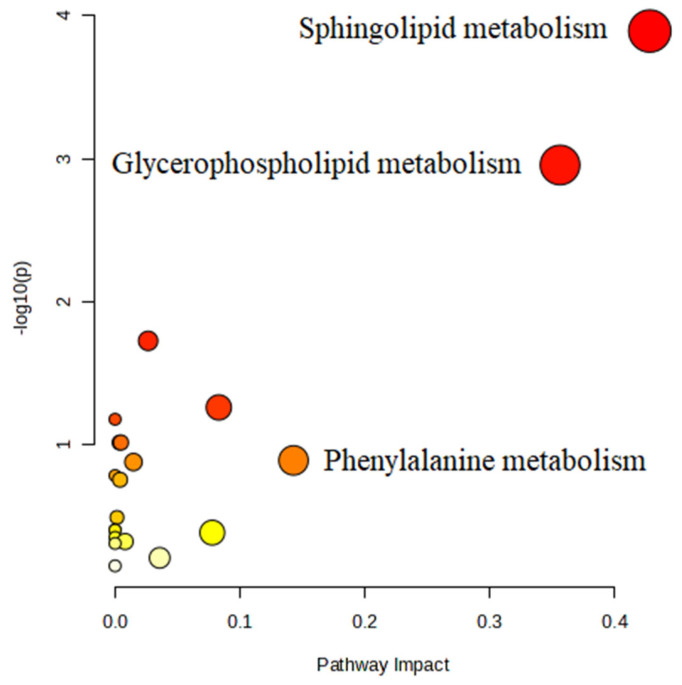
Metabolic pathways screened from KEGG library.

**Table 1 foods-13-00174-t001:** Seventy-nine differential compounds containing KEGG IDs.

Compound	RT (min)	VIP	*p*	MAX FC	KEGG ID
7-Dehydrocholesterol	16.94	1.0845	1.54 × 10^−6^	0.10403	C01164
Adenosine monophosphate	0.94	1.0692	5.07 × 10^−3^	0.14610	C00020
Glycocholic acid	8.21	1.0486	3.81 × 10^−4^	0.09227	C01921
Hypoxanthine	1.60	1.2362	1.55 × 10^−10^	2.61580	C00262
L-Proline	13.02	1.2902	6.60 × 10^−11^	2.42194	C00148
Inosine	1.60	1.1783	1.13 × 10^−5^	4.14642	C00294
Sphinganine	10.63	1.0807	1.87 × 10^−7^	7.40301	C00836
3-Methylindole	8.06	1.2503	6.84 × 10^−5^	0.43736	C08313
Undecanoic acid	10.19	1.4279	1.43 × 10^−10^	3.56285	C17715
Enkephalin L	16.51	1.1233	1.98 × 10^−3^	0.24478	C16041
N2-Succinyl-L-glutamic acid 5-semialdehyde	9.37	1.3988	3.65 × 10^−9^	2.13017	C05932
N1-Acetylspermidine	14.68	1.2693	1.02 × 10^−3^	3.82457	C00612
Desmosterol	17.63	1.3727	1.88 × 10^−6^	0.13485	C01802
LysoPC(18:1(9Z)/0:0)	9.80	1.1244	4.38 × 10^−7^	3.44659	C04230
Testosterone glucuronide	14.80	1.4248	5.75 × 10^−6^	0.12439	C11134
Cinnamaldehyde	8.59	1.6872	2.64 × 10^−6^	2.56298	C00903
Tryptophol	10.30	1.1046	2.51 × 10^−5^	0.27433	C00955
(2E)-Tetradecenoyl-CoA	15.52	1.0385	1.37 × 10^−2^	0.18103	C05273
Phytosphingosine	7.32	1.2752	1.55 × 10^−3^	3.46050	C12144
9,10-DHOME	6.45	1.8056	3.35 × 10^−8^	3.22173	C14828
FAPy-adenine	0.94	1.0645	4.08 × 10^−3^	0.11126	C06502
Cer(d18:1/22:0)	17.65	1.2753	1.59 × 10^−4^	0.00019	C00195
Cer(d18:1/24:0)	15.27	1.1066	5.44 × 10^−6^	0.00377	C00195
TG(16:0/16:0/18:1(9Z))	17.62	1.3167	7.99 × 10^−13^	0.13895	C00422
TG(16:0/18:0/18:2(9Z,12Z))	15.27	1.3851	1.30 × 10^−16^	0.20244	C00422
Phenylacetaldehyde	8.58	1.6699	2.66 × 10^−7^	5.34375	C00601
LysoPA(0:0/18:0)	9.69	1.2091	5.74 × 10^−7^	3.43180	C00416
PC(16:0/20:1(11Z))	13.03	1.3313	9.95 × 10^−8^	0.06575	C00157
PC(18:0/16:0)	13.04	1.4029	3.86 × 10^−6^	0.24483	C00157
PC(18:2(9Z,12Z)/16:0)	11.76	1.2954	1.67 × 10^−4^	0.07218	C00157
PC(18:3(6Z,9Z,12Z)/22:6(4Z,7Z,10Z,13Z,16Z,19Z))	12.01	1.2578	7.08 × 10^−4^	0.03811	C00157
PC(22:6(4Z,7Z,10Z,13Z,16Z,19Z)/20:4(5Z,8Z,11Z,14Z))	4.45	1.1355	1.54 × 10^−6^	0.33359	C00157
PE(14:0/24:0)	4.90	1.2762	1.05 × 10^−8^	0.39162	C00350
PE(14:1(9Z)/22:0)	12.22	1.1505	1.62 × 10^−6^	5.60701	C00350
PE(15:0/22:0)	11.76	1.2211	5.31 × 10^−9^	2.92555	C00350
PE(15:0/22:1(13Z))	11.76	1.2579	1.31 × 10^−8^	3.48521	C00350
PI(16:0/18:0)	18.12	1.3937	1.07 × 10^−11^	0.03992	C00626
PI(16:0/20:2(11Z,14Z))	18.12	1.3449	1.72 × 10^−12^	0.00061	C00626
PI(16:0/22:4(10Z,13Z,16Z,19Z))	18.12	1.3288	1.47×10^−11^	0.01661	C00626
PI(18:0/22:4(10Z,13Z,16Z,19Z))	15.48	1.2545	1.99 × 10^−9^	0.00180	C00626
LysoPC(16:0/0:0)	9.52	1.2091	6.29 × 10^−9^	3.27169	C04230
LysoPC(18:0/0:0)	10.85	1.0605	1.66 × 10^−4^	2.02727	C04230
LysoPC(18:2(9Z,12Z)/0:0)	8.95	1.0389	1.01 × 10^−4^	2.10668	C04230
LysoPC(20:0/0:0)	4.73	1.2112	9.47 × 10^−5^	0.18359	C04230
LysoPC(P-16:0/0:0)	9.85	1.1504	3.33 × 10^−6^	3.97018	C04230
4-(Methylnitrosamino)-1-(3-pyridyl)-1-butanone	13.63	1.2845	3.09 × 10^−9^	3.96652	C16453
SM(d18:1/24:1(15Z))	12.62	1.4618	4.69 × 10^−13^	0.09573	C00550
Histidinal	13.30	1.3567	4.60 × 10^−13^	3.15894	C01929
Methylarsonite	18.21	1.0455	6.45 × 10^−3^	0.49109	C07295
Acetamidopropanal	13.11	1.2957	3.63 × 10^−11^	2.23829	C18170
Monoethylhexyl phthalic acid	10.32	1.3609	1.33 × 10^−11^	2.44375	C03343
Tryptophanamide	8.18	1.7118	2.61 × 10^−5^	0.44664	C00977
SM(d18:1/26:1(17Z))	16.34	1.2566	3.94 × 10^−7^	0.05409	C00550
Azithromycin	11.96	1.3466	5.50 × 10^−7^	5.44225	C06838
Ethambutol	12.90	1.3167	1.24 × 10^−12^	3.11991	C06984
Carteolol	9.69	1.0615	1.46 × 10^−8^	5.10115	C06874
Nystatin	12.40	1.1696	6.16 × 10^−10^	0.02798	C06572
Proparacaine	10.23	1.3057	1.84 × 10^−13^	13.09740	C07383
Azlocillin	11.71	1.2460	5.78 × 10^−10^	4.77165	C06839
Procarbazine	11.08	1.2308	3.10 × 10^−4^	0.02835	C07402
Cinnamyl alcohol	10.81	1.2869	2.36 × 10^−5^	0.49156	C02394
Triethanolamine	13.39	1.3102	2.59 × 10^−12^	2.33451	C06771
Tomatidine	10.01	2.0814	1.90 × 10^−7^	2.11146	C10826
Sclareol	18.12	1.2857	5.30 × 10^−10^	0.00866	C09183
Pentabromodiphenyl ethers	14.69	1.3333	1.30 × 10^−5^	0.00016	C18203
Petanin	10.99	1.2599	3.45 × 10^−7^	0.10535	C12139
Resiniferatoxin	13.26	1.0417	1.11 × 10^−7^	2.28031	C09179
Oleandomycin	11.55	2.2712	1.82 × 10^−7^	4.94682	C01946
TETRATHIONIC ACID	18.17	1.1311	1.21 × 10^−3^	0.15010	C05529
Sodium palmitate	10.15	1.2995	2.83 × 10^−10^	2.67924	C00249
Octanoate	7.63	1.2354	3.20 × 10^−9^	0.29379	C06423
Gum arabic	9.24	1.2287	9.46 × 10^−4^	0	C08822
n-decanohydroxamic acid	12.63	1.3181	4.76 × 10^−10^	2.63773	C12889
Valproic acid	10.48	1.4104	3.14 × 10^−9^	2.27628	C07185
alpha,alpha′-Trehalose 6-mycolate	11.73	1.0273	8.41 × 10^−4^	0.04322	C04218
DG(O-16:0/18:1(9Z))	13.66	1.1011	2.27 × 10^−3^	2.11238	C13862
PC(22:0/22:1(13Z))	12.14	1.1250	1.32 × 10^−3^	0.30577	C00157
PE(18:4(6Z,9Z,12Z,15Z)/18:4(6Z,9Z,12Z,15Z))	4.64	1.2228	3.99 × 10^−8^	0.20370	C00350
PI(16:0/20:3(8Z,11Z,14Z))	14.72	1.2396	5.62 × 10^−6^	0.27455	C00626

**Table 2 foods-13-00174-t002:** List of metabolic pathways determined using enriched analysis for the metabolites.

Pathway Name	Match Status	*p*	−log (*p*)	Holm *p*	FDR	Impact
Sphingolipid metabolism	4/21	0.00013	3.89060	0.01081	0.01081	0.42800
Glycerophospholipid metabolism	4/36	0.00111	2.95530	0.09200	0.04655	0.35618
Glycerolipid metabolism	2/16	0.01872	1.72770	1	0.52412	0.02648
Purine metabolism	3/65	0.05457	1.26300	1	1	0.08302
Linoleic acid metabolism	1/5	0.06601	1.18040	1	1	0
Fatty acid elongation	2/39	0.09596	1.01790	1	1	0.00362
Fatty acid degradation	2/39	0.09596	1.01790	1	1	0.00467
Phenylalanine metabolism	1/10	0.12786	0.89325	1	1	0.14286
Fatty acid biosynthesis	2/47	0.13130	0.88175	1	1	0.01473
alpha-Linolenic acid metabolism	1/13	0.16308	0.78760	1	1	0
Glycosylphosphatidylinositol (GPI)-anchor biosynthesis	1/14	0.17452	0.75817	1	1	0.00399
Phosphatidylinositol signaling system	1/28	0.31978	0.49515	1	1	0.00152
Arachidonic acid metabolism	1/36	0.39149	0.40728	1	1	0
Biosynthesis of unsaturated fatty acids	1/36	0.39149	0.40728	1	1	0
Arginine and proline metabolism	1/38	0.40826	0.38906	1	1	0.07780
Steroid biosynthesis	1/42	0.44049	0.35607	1	1	0
Primary bile acid biosynthesis	1/46	0.47104	0.32694	1	1	0.00805
Aminoacyl-tRNA biosynthesis	1/48	0.48571	0.31362	1	1	0
Metabolism of xenobiotics by cytochrome P450	1/68	0.61263	0.21280	1	1	0.03571
Steroid hormone biosynthesis	1/85	0.69649	0.15709	1	1	0

## Data Availability

Data are contained within the article.

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
