# Peer review of "Changes in Meat of Hu Sheep during Postmortem Aging Based on ACQUITY UPLC I-Class Plus/VION IMS QTof"

_foods, 2024, doi:10.3390/foods13010174_

Round 1
Reviewer 1 Report
Comments and Suggestions for Authors
Title: Changes in meat of Hu sheep during postmortem aging based on ACQUITY UPLC I-Class Plus IMS QTof
The manuscript “Changes in meat of Hu sheep during postmortem aging based on ACQUITY UPLC I-Class Plus IMS QTof” investigated the effects of postmortem aging on Hu sheep meat quality in longissimus dorsi (LD) muscles during postmortem aging. Study shows the changes in basic meat quality, muscle fiber morphology and structure, as well as the relative changes in metabolomics and metabolic pathways during the postmortem maturation process of Hu sheep meat. Furthermore, Sphingolipid metabolism, Glycerophospholipid metabolism, Phenylalanine metabolism, Fatty acid elongation and degradation were observed to be significant metabolic pathways during postmortem ageing. It is well written article with some interesting findings; however, there are some corrections:
First objection is why authors didn’t include the line numbers as without them it is not possible for the reviewers to give suggestions?
Please add a concluding sentence at the end of the abstract.
In introduction part authors should add a sentence about usage of ACQUITY UPLC I-Class Plus IMS QTof in food industry and how its superior than other instruments used.
First sentence of the materials and methods: a total of 8 August old Hu sheep rams? What does this mean? Please rephrase the sentence to make it clear for the readers.
Within 30 minutes after slaughter, the longest dorsal muscle was taken…? Why authors separated the muscles without giving proper time of postmortem to the muscles with carcass? As commercially carcass is cut down after giving proper postmortem time?
Collect meat samples at a total of 11 time points from 0 to 168, and use a portable pH meter….? What is 0 and 168 here?
Add specifications of the pH meter?
It would be better to write the formula of the meat loss rate?
Write down the specifications of the Minolta meter?
Specifications and reference of using the c Kjeldahl nitrogen analyzer?
Authors should draw a layout of the Staining and Observation of Skeleton Proteins using HE stain in form of figure that may be included as supplementary file.
One of the major concerns with the study is why authors used 168h or 7 days as maximum time frame? As meat may be stored for 7days at 4 C without any quality degradation, therefore, authors should go beyond that time frame like to 14 days in order to measure the quality at some maximum time limits.
Italicize the longissimus dorsi at all places of the article.
Tenderness was increased at 12 h postmortem? Why authors ignored 24h time frame? As in many studies it is considered as the time when rigor mortis is achieved to full extent in lamb muscles? Therefore, it is an important time frame.
Authors should mention the significance among different time points of a specific parameter in the figure(s) 1-7.
How the PLS-DA models were validated? Did authors perform permutation test, multiple correlation coefficient (R2), cross-validated R2 (Q2) for that purpose? Mention in the text portion of the manuscript.
How the enrichment level of each KEGG metabolomic pathway was analyzed and what was the significance level?
The discussion section is very weak. The authors need to explain how differential metabolites can influence formation and resolution of rigor mortis? Therefore, it is need to include more reference studies and discuss each important compound identified.
Authors should rewrite the conclusion part with its chances of practical implication and futuristic vision i.e., in future where the scientists should focus.
English grammar and sentence structure should be revised and corrected throughout the manuscript. At some places, sentences are so long that they are difficult to understand.

English grammar and sentence structure should be revised and corrected throughout the manuscript. At some places, sentences are so long that they are difficult to understand.
Author Response
Dear Reviewer
Thank you very much for your comments and professional advice. These opinions help to improve our article. Based on your suggestion and request, we have made corrected modifications on the manuscript. Meanwhile, the manuscript had be reviewed and edited by language services of MDPI. We hope that our work can be improved again. Furthermore, we would like to show the details as follows:
Comment 1: First objection is why authors didn’t include the line numbers as without them it is not possible for the reviewers to give suggestions?
RE: Thanks for the comments on this study. I'm sorry for any inconvenience caused by my negligence. The line number has been added to the article.
Comment 2: Please add a concluding sentence at the end of the abstract.
RE: Thank you very much for your suggestion. At the end of the abstract, we added a concluding sentence, line numbers are 20-22.
Comment 3: In introduction part authors should add a sentence about usage of ACQUITY UPLC I-Class Plus IMS QTof in food industry and how its superior than other instruments used.
RE: Thank you very much for your suggestion. In introduction part, we added some sentences about usage of ACQUITY UPLC I-Class Plus IMS QTof, line numbers are 57-61.
Comment 4: First sentence of the materials and methods: a total of 8 August old Hu sheep rams? What does this mean? Please rephrase the sentence to make it clear for the readers.
RE: Thanks for the comments on this study. We have made modifications to the original text, line numbers are 69-70.
Comment 5: Within 30 minutes after slaughter, the longest dorsal muscle was taken…? Why authors separated the muscles without giving proper time of postmortem to the muscles with carcass? As commercially carcass is cut down after giving proper postmortem time?
RE: I'm sorry, I didn't understand your meaning very well. The slaughter of lambs is carried out after quarantine and complies with regulations.
Comment 6: Collect meat samples at a total of 11 time points from 0 to 168, and use a portable pH meter….? What is 0 and 168 here?
RE: Thanks for the comments on this study. I'm sorry for any inconvenience caused by my negligence. The content here should be from 0 to 168 h, which has been modified in the text, line numbers are 81-82.
Comment 7: Add specifications of the pH meter?
RE: Thank you very much for your suggestion. Specifications of the pH meter has been added in text, line numbers are 82-83.
Comment 8: It would be better to write the formula of the meat loss rate?
RE: Thank you very much for your suggestion. Formula of the meat loss rate has been added in text, line numbers are 105-106.
Comment 9: Write down the specifications of the Minolta meter?
RE: Thank you very much for your suggestion. The specifications of the Minolta meter have been added in text, line numbers are 110.
Comment 10: Specifications and reference of using the c Kjeldahl nitrogen analyzer?
RE: Thank you very much for your suggestion. Specifications and reference have been added in text, line numbers are 118-120.
Comment 11: Authors should draw a layout of the Staining and Observation of Skeleton Proteins using HE stain in form of figure that may be included as supplementary file.
RE: Thank you very much for your suggestion. But I'm sorry, I didn't understand your meaning very well. The HE staining in the text uses the cross-section of muscle fibers. Staying of Skeleton Proteins uses the longitudinal section of muscle fibers. On the one hand, both cannot be displayed simultaneously, and on the other hand, HE staining cannot act on skeletal proteins.
Comment 12: One of the major concerns with the study is why authors used 168h or 7 days as maximum time frame? As meat may be stored for 7days at 4 C without any quality degradation, therefore, authors should go beyond that time frame like to 14 days in order to measure the quality at some maximum time limits.
RE: Thanks for the comments on this study. We conducted a preliminary experiment to determine the experimental time. The experiment found that starting from 96h or 120h, the TVB-N of some meat samples began to exceed 15mg/100g. Therefore, considering the quality of the meat, a maximum time frame of 7 days was chosen.
Comment 13: Italicize the longissimus dorsi at all places of the article.
RE: Thank you very much for your suggestion. The format of longissimus has been modified. Line numbers of them are 9、23 and 63.
Comment 14: Tenderness was increased at 12 h postmortem? Why authors ignored 24h time frame? As in many studies it is considered as the time when rigor mortis is achieved to full extent in lamb muscles? Therefore, it is an important time frame.
RE: Thank you very much for your suggestion. Some relevant discussions have been added in the article, line numbers are 197-200.
Comment 15: Authors should mention the significance among different time points of a specific parameter in the figure(s) 1-7.
RE: Thank you very much for your suggestion. We have tried our best to make the modifications, but they are still a bit unsightly. You can provide valuable feedback on this again, and I will make further revisions based on your feedback.
Comment 16: How the PLS-DA models were validated? Did authors perform permutation test, multiple correlation coefficient (R2), cross-validated R2 (Q2) for that purpose? Mention in the text portion of the manuscript.
RE: Thanks for the comments on this study. For PLS-DA, we conducted a permutation test. The line numbers for the relevant content in the article are 313-322.
Comment 17: How the enrichment level of each KEGG metabolomic pathway was analyzed and what was the significance level?
RE: Thanks for the comments on this study. We use the Pathway analysis in metabananalysis 5.0 for analysis, with Enrichment method selecting Hypergeometric Test and Topology analysis selecting Relative betweenness Centrality.
Comment 18: The discussion section is very weak. The authors need to explain how differential metabolites can influence formation and resolution of rigor mortis? Therefore, it is need to include more reference studies and discuss each important compound identified.
RE: Thanks for the comments on this study. We added 3.7 for discussion on the impact of important metabolites on meat quality. The line numbers for the relevant content in the article are 351-395.
Comment 19: Authors should rewrite the conclusion part with its chances of practical implication and futuristic vision i.e., in future where the scientists should focus.
RE: Thank you very much for your suggestion. In the conclusion section, we added several key metabolites and their potential important roles in meat maturation. The line numbers for the relevant content in the article are 402-405.
Comment 20: English grammar and sentence structure should be revised and corrected throughout the manuscript. At some places, sentences are so long that they are difficult to understand.
RE: Thank you very much for your suggestion. The manuscript had be reviewed and edited by language services of MDPI. English Editing ID is english-74581.
Thank you very much for your attention and time. Look forward to hearing from you.

Reviewer 2 Report
Comments and Suggestions for Authors
The aim of this study is to investigate the changes in basal meat quality and metabolites of longissimus dorsi muscle of Hu sheep during postmortem maturation, and to identify the appropriate maturation time, metabolites and pathways of Hu sheep muscle . The title of the work has aroused a lot of interest but from the text the interest has gradually waned and very little is said about the part linked to the focus of the work.
Section: Introduction
The article focuses attention on the study of the longissimus dorsi (LD) of Hu sheep, which as the authors report, is a prized, high-quality meat widely consumed by the Chinese. In this section there is no information regarding the Hu sheep nor the qualitative characteristics of the meat of this breed. I believe that the introduction is not focused and does not give emphasis to the metabolic products during the post-mortem maturation process. A recommendation: any acronym used must be expressed in full at least the first time.
Section: Materials and methods
2.1. Sample preparation: it would be better to report it as “Animals” or “Experimental design”. Report the number, weight and age at slaughter as well as the sex of Hui sheep used for testing.
2.: the way of reporting the methods is not suitable for a high-level scientific journal like Foods. I suggest authors view and take the reference given as a writing example.
2.2. pH Explain at what time the pH was measured after sectioning.
2.2.2. Shear Forces It is unclear whether shear forces were measured on the raw and/or cooked meat sample. Why did the authors measure shear forces on cooked rather than raw samples? Explain and bring back into the discussion. Specify at which times the shear forces were measured.
2.2.3. Loss rate Specify whether on cooked or raw meat. Does it report how the result is expressed?
2.2.4. Color measurements Needs to be completely rewritten. At what time after cutting were the colorimetric parameters measured? Specify which illuminant was used, what the aperture diameter of the lens was, and which color scale was used. Write the color parameters a*, b*, L* in full.
2.2.5. TVB-N. Write in the complete report how the result is expressed. How many repetitions were done?
Section 2.3 Provide bibliographical references.
3. Results and discussions
It is advisable not to start the sentence with "with....", but to reformulate it;
3.2. color;3.3. muscle... and more: capital letter; Please use "may be" instead of "may be", especially in the section of metabolites that have no bibliographic support to confirm the data.
3.1. pH and sharing forces 3.2. color, loss rate and TVB-N. Nothing new. Too long discussion. It's more like a review!
In report 3.2 the author (“Effect of animal age, post-mortem cooling rate and aging time on meat quality attributes of water buffalo and humped cattle bulls 2020”) .
I'm surprised that there are no bibliographical references (extensive bibliography on the topic) to strengthen the discussion.
The discussion on metabolites is very interesting but I don't find that there are sufficient arguments to understand the correlations between aging and metabolites and therefore connect them to the quality of meat.
It would have been very interesting to evaluate the correlation between the metabolites found, TVB-N and pH values.
I advise authors to pay more attention to the preparation of the manuscript: the font must always be the same.
Author Response
Dear Reviewer
Thank you very much for your comments and professional advice. These opinions help to improve our article. Based on your suggestion and request, we have made corrected modifications on the manuscript. Meanwhile, the manuscript had be reviewed and edited by language services of MDPI. We hope that our work can be improved again. Furthermore, we would like to show the details as follows:
Comment 1: The article focuses attention on the study of the longissimus dorsi (LD) of Hu sheep, which as the authors report, is a prized, high-quality meat widely consumed by the Chinese. In this section there is no information regarding the Hu sheep nor the qualitative characteristics of the meat of this breed. I believe that the introduction is not focused and does not give emphasis to the metabolic products during the post-mortem maturation process. A recommendation: any acronym used must be expressed in full at least the first time.
RE: Thanks for the comments on this study. We have added some introductions to the meat quality of Hu sheep in the article. Line numbers are 50-54.
Comment 2: 2.1. Sample preparation: it would be better to report it as “Animals” or “Experimental design”. Report the number, weight and age at slaughter as well as the sex of Hui sheep used for testing.
RE: Thank you very much for your suggestion. "Sample preparation" has been changed to "Animals". The number, weight, age, and gender at the time of slaughter have increased in section 2.1. Line numbers are 68-70.
Comment 3: 2.: the way of reporting the methods is not suitable for a high-level scientific journal like Foods. I suggest authors view and take the reference given as a writing example.
RE: Thank you very much for your suggestion. The way of reporting the methods has been modified, line numbers are 70-77.
Comment 4: 2.2. pH Explain at what time the pH was measured after sectioning.
RE: Thank you very much for your suggestion. Relevant content has been added to the article, line numbers are 81-82.
Comment 5: 2.2.2. Shear Forces It is unclear whether shear forces were measured on the raw and/or cooked meat sample. Why did the authors measure shear forces on cooked rather than raw samples? Explain and bring back into the discussion. Specify at which times the shear forces were measured.
RE: Thanks for the comments on this study. We used cooked meat. On the one hand, we conducted experiments based on references(line numbers are 95), and on the other hand, we found through previous preliminary experiments that using cooked meat for shear force measurement resulted in small differences between replicates at the same time, while for raw meat, the opposite was true. The measurement time of shear force has been added to the text, line numbers are 90-91.
Comment 6: 2.2.3. Loss rate Specify whether on cooked or raw meat. Does it report how the result is expressed?
RE: Thanks for the comments on this study. We used raw meat. The expression of the results we have added in the text, line numbers are 104-106.
Comment 7: 2.2.4. Color measurements Needs to be completely rewritten. At what time after cutting were the colorimetric parameters measured? Specify which illuminant was used, what the aperture diameter of the lens was, and which color scale was used. Write the color parameters a*, b*, L* in full.
RE: Thanks for the comments on this study. The time for measuring the color has been added to the text, line numbers are 116-117. The measurement aperture of the CR400 is 8mm, the light source is C, and the color difference reference color is 100. These contents have been added to the text, line numbers are 110-112.
Comment 8: 2.2.5. TVB-N. Write in the complete report how the result is expressed. How many repetitions were done?
RE: Thanks for the comments on this study. We have made modifications in the text, line numbers are 126-135.
Comment 9: Section 2.3 Provide bibliographical references.
RE: Thank you very much for your suggestion. The references have been added in the text, line numbers are 138 and 144-145.
Comment 10: It is advisable not to start the sentence with "with....", but to reformulate it;
RE: Thank you very much for your suggestion. We have made modifications in the text, line numbers are 175-176.
Comment 11: 3.2. color;3.3. muscle... and more: capital letter; Please use "may be" instead of "may be", especially in the section of metabolites that have no bibliographic support to confirm the data.
RE: Thank you very much for your suggestion. The format of the title has been modified, line numbers are 222 and 268. May be has been replaced with maybe.
Comment 12: 3.2. color;3.3. muscle... and more: capital letter; Please use "may be" instead of "may be", especially in the section of metabolites that have no bibliographic support to confirm the data.
RE: Thanks for the comments on this study. We have made certain modifications to these two parts.
Comment 13: In report 3.2 the author (“Effect of animal age, post-mortem cooling rate and aging time on meat quality attributes of water buffalo and humped cattle bulls 2020”) .
RE: Thank you very much for your suggestion. We have made modifications to the reference format here, line numbers are 223-224.
Comment 14: I'm surprised that there are no bibliographical references (extensive bibliography on the topic) to strengthen the discussion.
RE: Thanks for the comments on this study. I'm sorry for any inconvenience caused by my negligence. We added 3.7 for discussion on the impact of important metabolites on meat quality. The line numbers for the relevant content in the article are 351-395.
Comment 15: The discussion on metabolites is very interesting but I don't find that there are sufficient arguments to understand the correlations between aging and metabolites and therefore connect them to the quality of meat.
RE: Thanks for the comments on this study. In section 3.7, we discussed the key metabolites involved in the changes and elucidated their potential impact on meat quality.
Comment 16: It would have been very interesting to evaluate the correlation between the metabolites found, TVB-N and pH values.
RE: Thank you very much for your suggestion. We have added relevant discussions according to your feedback.
Comment 17: I advise authors to pay more attention to the preparation of the manuscript: the font must always be the same.
RE: Thank you very much for your suggestion. We have made unified modifications to the font format throughout the entire text.
Thank you very much for your attention and time. Look forward to hearing from you.

Reviewer 3 Report
Comments and Suggestions for Authors
The article with the title "Changes in meat of Hu sheep during postmortem aging based on UPLC-Q-TOFMS" has 15 pages with 16 references, 3 tables and 13 figures.
The article deals with a topic that is still relevant and the results will be interesting for the authors of other studies as well as for the evaluation of currently achieved quality parameters.
I recommend the author to edit the manuscript in accordance with the instructions to the authors, especially in the case of font size and style. And likewise in the case of references. Follow the instructions at https://mdpi-res.com/data/mdpi_references_guide_v5.pdf
Introduction
Also, I recommend expanding the number of references, especially for Material and methods and Results and discussions chapters, since most of the references are here in the introduction chapter.
Material and methods
In the text below (+), since the authors do not specify line spacing and I cannot mark the place in the text, the facts are presented in a superficial way. My critical assessment is also due to the nature of the study, as they deal with qualitative parameters of the postmortem kind. Thus, the method of slaughter, stunning, handling before slaughter and other information are essential in evaluating meat quality. So I recommend that the authors look at similar studies and add information. Facts regarding transport, nutritional status, etc. are usually listed there.
(+) The selected sheep were fasted for 24 hours before slaughter, and after 2 hours of water deprivation, they were collectively slaughtered. Within 30 minutes after slaughter, the longest dorsal muscle was taken from the posterior edge of the last rib at a distance of 3cm from the midline of the back, vacuum packaged, and refrigerated at 0-4 ℃.
There are also formal flaws in the text.
E.g. here:
The pH value was determined according to the method of Huang, Liu, et al. (2022) with slight modifications. = extra spaces in the text.
In the text below (++), is the copied text of the instructions for the pH meter? This is unacceptable.
(++) Clean the probe with ultrapure water and calibrate the pH meter before measurement; When measuring, make the probe completely in contact with the meat sample, avoiding fat and tendons. Wait for the reading to stabilize before counting. Repeat the measurement for each group of meat samples no less than 3 times; After each measurement, rinse the probe with ultrapure water and calibrate it.
In the text below (+++), is the copied text of the instructions for the texture measurement? This is unacceptable.
(+++) Take meat samples at different time points, remove surface fat, tendons, and membranes, place the meat samples in a polyethylene bag, heat them in a 75 ° C water bath until the center temperature of the meat sample is 70 ° C, and then take them out and let them cool to room temperature. Use a circular sampler equipped with a tenderness meter to drill along the direction parallel to the muscle fibers of the meat sample. The sampling position should be no less than 5mm away from the edge of the sample, and the sampling interval should be no less than 5mm. Use a meat tenderness tester(RH-N50; Guangdong, China) to measure the shear force value of the taken samples. The number of samples should be no less than 6, and the average value should be taken after removing abnormal values.
And I could go on like this. A redesign of the trial methodology is necessary to respect the publication requirements of this article.
Chapter number 3. (Results and discussion) is more or less without discussion. There are too many errors and typos, and the authors use semicolons instead of standard sentence endings.
Although there is a statistical evaluation using the P value in table number 3, the number of decimal places is non-standard and the overall concept of expressing the results and discussion is outside the usual practices and does not meet the standards for publishing.
Overall Recommendation is: Accept after minor/major revision (corrections to MAJOR methodological errors and text editing)
Author Response
Dear Reviewer
Thank you very much for your comments and professional advice. These opinions help to improve our article. Based on your suggestion and request, we have made corrected modifications on the manuscript. Meanwhile, the manuscript had be reviewed and edited by language services of MDPI. We hope that our work can be improved again. Furthermore, we would like to show the details as follows:
Comment 1: I recommend the author to edit the manuscript in accordance with the instructions to the authors, especially in the case of font size and style. And likewise in the case of references. Follow the instructions at https://mdpi-res.com/data/mdpi_references_guide_v5.pdf.
RE: Thank you very much for your suggestion. We have made modifications to the reference format of the entire text.
Comment 2: Also, I recommend expanding the number of references, especially for Material and methods and Results and discussions chapters, since most of the references are here in the introduction chapter.
RE: Thank you very much for your suggestion. We have added a certain number of references in Materials and methods and Results and discussion chapters.
Comment 3: Material and methods
In the text below (+), since the authors do not specify line spacing and I cannot mark the place in the text, the facts are presented in a superficial way. My critical assessment is also due to the nature of the study, as they deal with qualitative parameters of the postmortem kind. Thus, the method of slaughter, stunning, handling before slaughter and other information are essential in evaluating meat quality. So I recommend that the authors look at similar studies and add information. Facts regarding transport, nutritional status, etc. are usually listed there.
(+) The selected sheep were fasted for 24 hours before slaughter, and after 2 hours of water deprivation, they were collectively slaughtered. Within 30 minutes after slaughter, the longest dorsal muscle was taken from the posterior edge of the last rib at a distance of 3cm from the midline of the back, vacuum packaged, and refrigerated at 0-4 ℃.
RE: Thank you very much for your suggestion. We have made modifications to the content involved, line numbers are 72-77.
Comment 4: There are also formal flaws in the text.
E.g. here:
The pH value was determined according to the method of Huang, Liu, et al. (2022) with slight modifications. = extra spaces in the text.
RE: Thanks for the comments on this study. I'm sorry for any inconvenience caused by my negligence. We have made modifications throughout the entire text regarding this type of issue.
Comment 5: In the text below (++), is the copied text of the instructions for the pH meter? This is unacceptable.
(++) Clean the probe with ultrapure water and calibrate the pH meter before measurement; When measuring, make the probe completely in contact with the meat sample, avoiding fat and tendons. Wait for the reading to stabilize before counting. Repeat the measurement for each group of meat samples no less than 3 times; After each measurement, rinse the probe with ultrapure water and calibrate it.
RE: Thanks for the comments on this study. We have made modifications to this, line numbers are 83-87.
Comment 6: In the text below (+++), is the copied text of the instructions for the texture measurement? This is unacceptable.
(+++) Take meat samples at different time points, remove surface fat, tendons, and membranes, place the meat samples in a polyethylene bag, heat them in a 75 ° C water bath until the center temperature of the meat sample is 70 ° C, and then take them out and let them cool to room temperature. Use a circular sampler equipped with a tenderness meter to drill along the direction parallel to the muscle fibers of the meat sample. The sampling position should be no less than 5mm away from the edge of the sample, and the sampling interval should be no less than 5mm. Use a meat tenderness tester(RH-N50; Guangdong, China) to measure the shear force value of the taken samples. The number of samples should be no less than 6, and the average value should be taken after removing abnormal values.
RE: Thanks for the comments on this study. We have made modifications to this, line numbers are 91-97.
Comment 7: And I could go on like this. A redesign of the trial methodology is necessary to respect the publication requirements of this article.
RE: Thanks for the comments on this study. I'm sorry for any inconvenience caused by my negligence. We have made certain modifications to the experimental methods in the article.
Comment 8: Chapter number 3. (Results and discussion) is more or less without discussion. There are too many errors and typos, and the authors use semicolons instead of standard sentence endings.
RE: Thanks for the comments on this study. We added 3.7 for further discussion, line numbers are 351-395. In addition, modifications have been made to errors and typos.
Comment 9: Although there is a statistical evaluation using the P value in table number 3, the number of decimal places is non-standard and the overall concept of expressing the results and discussion is outside the usual practices and does not meet the standards for publishing.
RE: Thank you very much for your suggestion. We have made unified modifications to the decimal places in the table. Then, according to the publishing requirements, certain modifications were made to the format of the entire text.
Thank you very much for your attention and time. Look forward to hearing from you.

Round 2
Reviewer 1 Report
Comments and Suggestions for Authors
The manuscript is sufficiently improved based on comments and suggestions of the reviewers.
Reviewer 2 Report
Comments and Suggestions for Authors
Congratulations. The paper can be accepted.